# The Role of Contrast-Enhanced Ultrasound in the Differential Diagnosis of Tuberous Vas Deferens Tuberculosis and Metastatic Inguinal Lymph Nodes

**DOI:** 10.3390/diagnostics13101762

**Published:** 2023-05-17

**Authors:** Wenzhi Zhang, Tu Ni, Wei Tang, Gaoyi Yang

**Affiliations:** Department of Ultrasonography, Affiliated Hangzhou Chest Hospital, Zhejiang University School of Medicine (Hangzhou Red Cross Hospital), Hangzhou 310003, China

**Keywords:** contrast-enhanced ultrasound, vas deferens, tuberculosis, metastatic, lymph nodes

## Abstract

Purpose: To retrospective analysis and summary the features of tuberous vas deferens tuberculosis (VD TB) and inguinal metastatic lymph nodes (MLN) in routine ultrasound (US) and contrast-enhanced US (CEUS) as well as to assess the utility of CEUS in differentiating between the two diseases. Methods: The US and CEUS findings of patients with pathologically confirmed tuberous VD TB (*n* = 17) and inguinal MLN (*n* = 28), including the number of lesions, presence of bilateral disease, differences in internal echogenicity, a conglomeration of lesions, and blood flow within the lesions, were retrospectively analyzed. Results: Routine US showed no significant difference in the number of lesions, nodule size, internal echogenicity, sinus tract, or skin rupture; however, significant differences were observed between the two conditions in the conglomeration of lesions (χ^2^ = 6.455; *p* = 0.023) and the degree, intensity, and echogenicity pattern on CEUS (χ^2^ = 18.865, 17.455, and 15.074, respectively; *p* = 0.000 for all). Conclusions: CEUS can show the blood supply of the lesion, and judge the physical condition of the lesion better than US. Homogeneous, centripetal, and diffuse enhancement should prompt a diagnosis of inguinal MLN, whereas lesions with heterogeneous and diffuse enhancement on CEUS should be considered as VD TB. CEUS has great diagnostic value in differentiating between tuberous VD TB and inguinal MLN.

## 1. Introduction

The vas deferens (VD) is a part of the male reproductive system. It is a direct continuation of the epididymal tube. It is about 50 cm long. The length of the vas deferens is 40–50 cm [1], from the end of the epididymis, along the posterior edge of the testis up into the spermatic cord. It passes through the spermatic cord and enters the abdominal cavity via the inguinal canal, immediately bending inward and descending into the pelvic cavity. Passing anteriorly above the end of the ureter to posteriorly behind the base of the bladder. Here, the vas deferens on both sides gradually approach and expand into the ampulla of the vas deferens. So the vas deferens are anatomically divided into the pelvic VD, groin VD, spermatic VD, and testes VD [2].

Vas deferens tuberculosis is a chronic mycobacterium tuberculosis infection of the vas deferens, which is mostly secondary to tuberculosis of the genitourinary system [3] and can also be caused by hematogenous infection. VD TB is one of the causes of male infertility [3,4]. According to US manifestation, VD TB can be divided into three types: uniformly thickened wall type, tuberous type, and abscess type [2]. Genital tuberculosis is uncommon [5], and the incidence of tuberous VD TB is also very low, which leads to the lack of understanding of imaging physicians and often leads to misdiagnosis. Metastatic inguinal lymph nodes are diseases caused by metastasis of malignant tumors from the drainage area of inguinal lymph nodes to lymph nodes in the groin. Common malignant tumors are: primary malignant tumors of the genitalia, skin, rectum, or prostate [6,7]. Misdiagnosis of metastatic lymph nodes often leads to adverse consequences, so early and accurate diagnosis of metastatic lymph nodes is very important. Many scholars have studied the imaging diagnosis of metastatic lymph nodes in the groin [8,9,10,11], and the positive rate of diagnosis is relatively high, The ultrasound manifestations of metastatic inguinal lymph nodes were: cortical thickening, medulla compression and deformation, disappearance of lymphatic hilum, sandy calcification, and lymph node fusion. The results of CEUS showed centripetal enhancement, peripheral enhancement, diffuse enhancement, and cycle-like enhancement. In recent years, with the rapid development of ultrasound technology, the resolution of the instrument has been very high. Due to the fixed anatomical position of the normal lymph nodes in the inguinal region, the superficial position, and clear structure, it is not easy to misdiagnose. The abnormal structure of inguinal metastatic lymph nodes, the disappearance of normal lymphatic hilum structure, is not easy to distinguish, it may be confused with other diseases, Tuberous type of VD TB in the inguinal region and spermatic division is easily misdiagnosed as metastatic lymph nodes, or misdiagnosed as tuberculosis of the inguinal region with peripheral abscess. The indications for CEUS have increased rapidly since the first clinical applications in the late 1990s [12]. At present, CEUS has been applied to the liver, kidney, thyroid, breast, and lymph nodes [13,14]. The application of CEUS in deferens tuberculosis is rarely reported. This study retrospectively analyzed the routine ultrasound and CEUS findings of tuberous VD TB and metastatic inguinal lymph nodes, providing a reference for the differential diagnosis of the two diseases. To our knowledge, this is the largest study of tuberous VD TB in the world.

## 2. Materials and Methods

### 2.1. Patients and Study Design

This study was reviewed and approved by the Medical Ethics Committee of Hangzhou Red Cross Hospital, and written informed consent was obtained from patients. All methods were performed in accordance with the relevant guidelines and regulations. Between March 2011 and March 2022, 17 cases of tuberous VD TB and 28 cases of inguinal MLN, which had been confirmed by pathology, were evaluated using US and CEUS at our facility. All VD TB patients were in the advanced stage of the disease, inguinal MLN patients are in the later stages of the disease. We only included patients who underwent US and CEUS, those with complete medical information, and those without severe cardiopulmonary dysfunction. Other inclusion criteria were patients diagnosed with tuberous VD TB or inguinal MLN who showed positive findings on US and CEUS of the inguinal lesions, Structural abnormalities, such as obvious thickening of the lymph cortex, disappearance of the lymphatic hilum, and imbalance of aspect ratio, showed diffuse enhancement in CEUS. Exclusion criteria included patients with contraindications for CEUS, those with intact lymph node structure and morphology, those with inguinal MLN in whom lymphatic hilum was clearly visible, those with hyperechoic (calcified) VD TB, and those with tuberous VD TB after surgical excision of epididymal tuberculosis.

### 2.2. US and CEUS Examination

A Philips ultrasonic diagnostic instrument (iU22, Philips Healthcare, Bothell, WA, USA) with a high-frequency linear array probe (L12-5, frequency 5–12 MHz; L9-3, frequency 3–9 MHz) and low-frequency convex array probe (C5-1, frequency 1–5 MHz) was used for imaging. Patients were placed in the supine position with a fully exposed scrotum and groin. First, we performed bilateral routine ultrasonic observation of the inguinal region with or without abnormal echogenicity lesions to assess their quantity, size, internal echoes, and relationship with the spermatic cord. Next, color Doppler flow imaging (CDFI) was used to evaluate (i) blood supply and continuity, (ii) nodules in the inguinal region and in the spermatic and testicular parts of the VD, and (iii) testis and epididymis for abnormal echogenicity. Subsequently, CDFI was used to evaluate the blood supply to the VD. Range of motion was determined by squeezing the lesion using the probe (If the lesion is relatively active with the surrounding tissue after the probe compression, that is, the lesion has good activity; if the lesion is not active with the surrounding tissue after the probe compression, or the lesion is fixed with the surrounding tissue, that is, the lesion has poor activity). The CEUS examination was performed using low mechanical index (0.06) pulse reverse harmonic imaging and a sulfur hexafluoride microbubble ultrasonic contrast agent SonoVue (Milan, Italy, Bracco SpA). Specifically, the elbow vein was injected with 2.4 mL, followed by 5 mL of saline. Further, dynamic observation of lesion enhancement was followed by continuous observation for 2 min. All images were stored in the instrument’s hard disk for subsequent analysis. To reduce subjective error, routine US and CEUS were performed by two attending physicians with 5 years of experience who were unaware of the results of the pathological analysis. All images were independently diagnosed and analyzed by the two sonographers, followed by a discussion to arrive at a consensus.

### 2.3. Statistical Analyses

We compared the US and CEUS findings between tuberous VD TB and inguinal MLN. All data were analyzed using SPSS ver. 23.0 statistical software (Armonk, New York, NY, USA, IBM). Numerical data and differences between the US and CEUS modes were analyzed using χ^2^ and Fisher accurate tests. *p* < 0.05 was defined as statistically significant.

## 3. Results

### 3.1. Patients

The tuberous VD TB group (17 cases) comprised 17 men aged between 19–52 years, with an average age of 29.87 ± 5.34 years. The inguinal MLN group (28 cases) included 28 men aged between 46–75 years, with an average age of 52.33 ± 4.71 years. Antituberculosis treatment led to a complete cure in 3 cases; thus, 14 cases were confirmed by pathology after surgery. All cases of inguinal MLN were pathologically confirmed by biopsy, and metastases were noted in 6 cases of intestinal tumor, 8 cases of prostate cancer, 7 cases of kidney cancer, 5 cases of melanoma, 1 case of testicular tumor, and 1 case of penile cancer.

### 3.2. US Examination

Data regarding the number of lesions, maximum nodule size, internal echogenicity, calcification foci, conglomeration of lesions, and sinus tract and skin rupture for both conditions are summarized in Table 1. Two patients with metastatic lymph nodes were thin, with large diseased lymph nodes and skin ulcers. The distribution of tuberous VD TB was linear (Figure 1), whereas that of inguinal MLN was nonlinear (Figure 2).

### 3.3. CEUS Examination

In patients with VD TB, enhancement on CEUS was homogeneous (3/17, 17.64%), heterogeneous (11/17, 64.7%), or showed no enhancement (3/17, 17.6%). As most images were highly enhanced, heterogeneous enhancement was categorized as septal (4/17, 23.5%) (Figure 3), annular (4/17, 23.5%), or nodule-in-nodule (3/17, 17.6%) enhancement. In this study, some VDTB showed low enhancement (Figure 4). All images showed diffuse enhancement. In patients with inguinal MLN, CEUS showed homogeneous (23/28, 82.1%) or heterogeneous (5/28, 17.8%) enhancement, and both were highly enhanced (28/28, 100%) (Figure 5). The main enhancement mode was centripetal enhancement (75.0%, 16/28), and noncentripetal enhancement referred to diffuse (25.0%, 10/28) or lymphatic hilar (7.1%, 2/28) enhancement in the periphery. Heterogeneously enhanced MLN was noted in 3 cases of rectal cancer, 1 case of prostate cancer, and 1 case of kidney cancer. All relevant data are shown in Table 2.

## 4. Discussion

The incidence of VD TB in extrapulmonary tuberculosis is extremely low, and many cases are secondary to testicular and epididymal TB and prostate TB. Given that VD TB is a chronic condition [3,15], it is inadvertently found mostly in patients with scrotal masses or scrotal pain and those who have a straining feeling, and in patients with a history of TB or who have typical symptoms of TB; the main pathological changes of caseous degeneration and necrosis [16,17,18], accompanied by the progress of the course, as shown by ultrasound vary [19]. Tuberculous VD is commonly observed in ultrasound examinations. Mycobacterium tuberculosis proliferates in the VD during disease progression, and the VD wall thickens and is then destroyed. Simultaneously, caseous substances block the VD lumen and form local nodules. Tuberculous VD mainly shows hypoechoic and mixed echogenicity signals, and rarely presents strong echogenicity [2]. Because a tuberculous pattern with strong echogenicity is usually observed in the later disease stage, it can be easily distinguished from the metastatic lymph nodes on the basis of internal echogenicity; therefore, patients with these patterns were not included in this study. Among the 17 patients with VD TB, hypoechoic tuberculous patterns accounted for 58.8% (10/17), and tuberculous patterns with mixed echogenicity accounted for 35.3% (6/17), and both patterns were observed in the progressive stage of TB. The anatomical locations of VD TB were similar to those of inguinal lymph nodes, which means that sonographers who lack an understanding of the disease can easily misdiagnose it.

The standard method to assess regional lymph node metastasis is surgical inguinal lymphadenectomy [20,21]. In recent years, sentinel lymph node biopsy has proven to be a reliable tool for assessing local lymph node status [22]. Because up to 70% of patients have groin or lower-limb problems after radical groin surgery, preoperative determination of lymph node-positive disease is a major concern [11,22,23,24]. Therefore, non-invasive CEUS has a potentially wide application. CEUS can enable preliminary evaluation of the morphology, structure, and blood supply of lymph nodes [25,26,27,28], provide a preliminary evaluation of lymph node biopsy, and guide lymph-node puncture biopsy. These benefits are crucial because misdiagnosis of lymph node metastasis can often lead to patient death [11]. The success rate of CEUS in metastatic inguinal lymph nodes has been reported to be 94.7%, which is comparable to that of conventional sentinel lymph node assessments (88.8% and 94.1%) [22]. However, the sensitivity of CEUS to VD TB needs to be further explored.

The ultrasound diagnostic criteria published by Abang showed a sensitivity of 89% for diagnosing the lymph node status. In these criteria, a short-axis diameter and long-axis/short-axis ratio were used to identify positive nodes [29]. In the present study, we performed a routine ultrasonic evaluation of the number of lesions, focal size, internal echogenicity, and internal calcifications reference index to distinguish between the two types of diseases; however, there were no significant differences except in the incidence of conglomeration, which was higher in patients with VD TB than in patients with metastatic inguinal lymph nodes. This finding was associated with the anatomical features and pathological physiology. Because of the long tubular structure in the VD, segmental lesions can occur, which are prone to so-called focal conglomeration (Figure 1). An interesting finding in this study was that VD TB showed a linear distribution, whereas the metastatic lymph nodes in the groin showed a non-linear distribution (Figure 2), which is a crucial ultrasound differential diagnosis index. The ultrasound findings described above are associated with the VD anatomy and distribution characteristics of inguinal lymph nodes.

The effectiveness of ultrasound in assessing inguinal lymph nodes has been demonstrated in other studies [30,31]. Inguinal lymph nodes are located close to the skin. Ultrasound has a high sensitivity to inguinal lymph nodes and the advantage of not exposing patients to ionizing radiation; thus, it is widely used in clinical inguinal lymph node examinations [10]. Metastatic inguinal lymph nodes are mostly observed in primary malignant tumors of the genital and reproductive organs, skin, rectum, anus, or bladder [32]. Inguinal lymph node metastasis is uncommon for malignancies above the diaphragm because the inguinal lymph nodes do not receive lymph from the lungs, and these metastases are caused by systemic vascular spread [22,33] or lymphatic drainage deviations caused by tumor-related obstructions [34]. The 5-year cancer-specific survival rate of patients without other diseases is 98.4%. In the presence of lymph node metastasis, this ratio considerably decreases [35]. Therefore, early detection and treatment can affect the prognosis of the disease.

In this study, the primary lesions of metastatic inguinal lymph nodes were found in the skin, gastrointestinal tract, reproductive organs, and urinary system. Metastatic lymph nodes mostly retain the biological characteristics of the primary lesion [36,37]. Pathologically poorly differentiated and rapidly growing tumors will metastasize to lymph nodes and easily lead to internal lymph node necrosis. In this study, five patients with metastatic inguinal lymph nodes showed annular enhancement by CEUS, and no enhancement area was observed inside. The ultrasonic imaging in the VD TB group showed a similar ring enhancement that was slightly thicker than the thickness of VD TB. These findings may be associated with the inflammatory lesions in VD TB in which the VD wall thickens and pus forms in the lumen simultaneously, which leads to tube cavity expansion as the pus accumulates, and as the pressure increases, the wall becomes thinner. This possible mechanism should be confirmed via further studies with a larger cohort of patients with VD TB. Among the 17 patients with VD TB in our study, 4 showed septal enhancement by CEUS (Figure 3), whereas those with metastatic lymph nodes did not. We believe that this was a characteristic ultrasonic manifestation of VD TB, and a greater septal enhancement by CEUS could enable physicians to exclude metastatic lymph nodes. Due to the limited sample size in this study, the view that separated sample enhancement is the characteristic CEUS manifestation of VD TB needs to be further confirmed by increasing the sample size.

Most metastatic lymph nodes showed homogeneous enhancement (82.14%, 23/28), and all were highly enhanced (Figure 4), whereas only 17.64% (3/17) of VD TB were heterogeneously enhanced, and the proportion of high enhancement was not high (47.05%, 8/17). It can be seen that in the case of a full understanding of the patient’s medical history. The lesions that showed homogeneous high enhancement tended to be diagnosed as MLN. In CEUS, the lesions that showed heterogeneous enhancement and low enhancement (Figure 5) tended to be diagnosed as VD TB.

In this study, there was a big difference in the mean age of the two groups of patients. VD TB is most likely to occur in young adults, while inguinal MLN tends to occur in middle and old age. Therefore, age is also an important factor in differential diagnosis.

There may be some possible limitations in this study. Firstly, it is a retrospective study, and its results must be confirmed by a large number of prospective studies. Secondly, this study is a single-center study with a small sample size, so the research results need to be further confirmed by multi-center studies and large-sample studies

## 5. Conclusions

CEUS can help clearly visualize blood supply in tuberous VD TB and inguinal MLN. Hence, it has important value in the differential diagnosis of these two conditions, which is based on the evaluation of the form, distribution characteristics, and internal echogenicity of the lesions via routine US or CEUS.

## Figures and Tables

**Figure 1 diagnostics-13-01762-f001:**
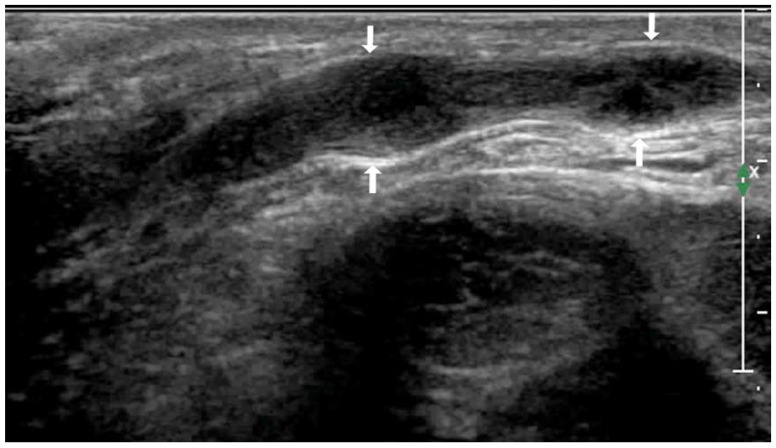
A 30-year-old male with tuberous VD TB (arrow). Ultrasound shows hypoechoic nodules (arrow) with conglomeration.

**Figure 2 diagnostics-13-01762-f002:**
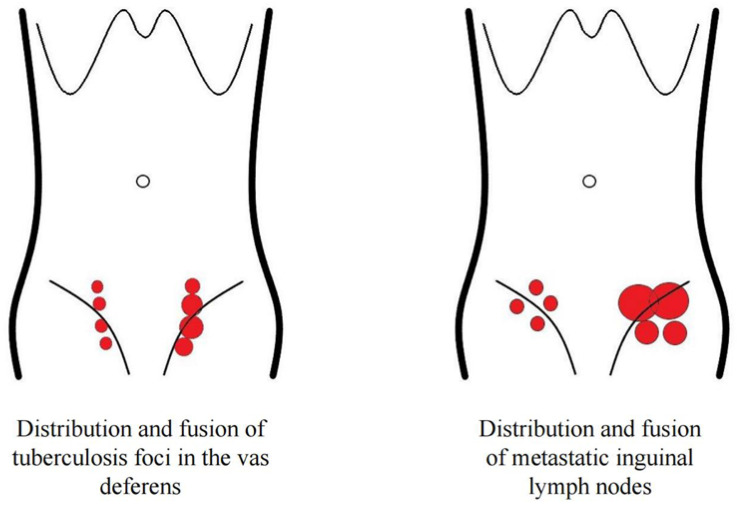
The focal distribution and progression of tuberous VD TB and the inguinal MLN.

**Figure 3 diagnostics-13-01762-f003:**
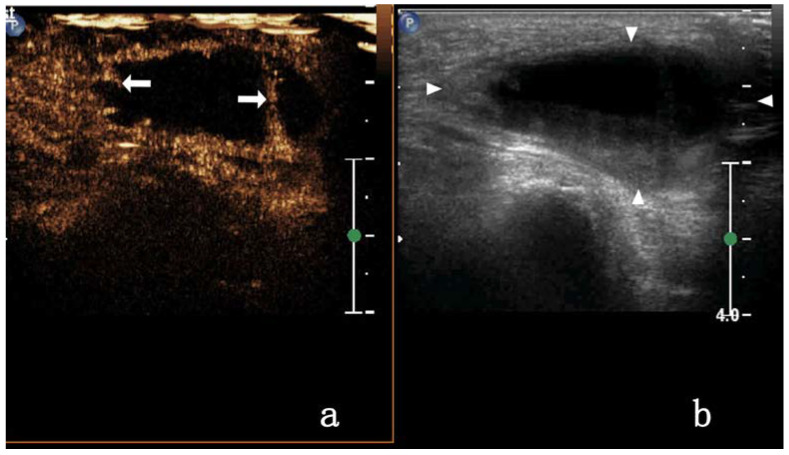
A 31-year-old male with tuberous VD TB. CEUS shows septal enhancement (arrow) in (**a**); 2D US presents mixed echo (triangular arrow) in (**b**).

**Figure 4 diagnostics-13-01762-f004:**
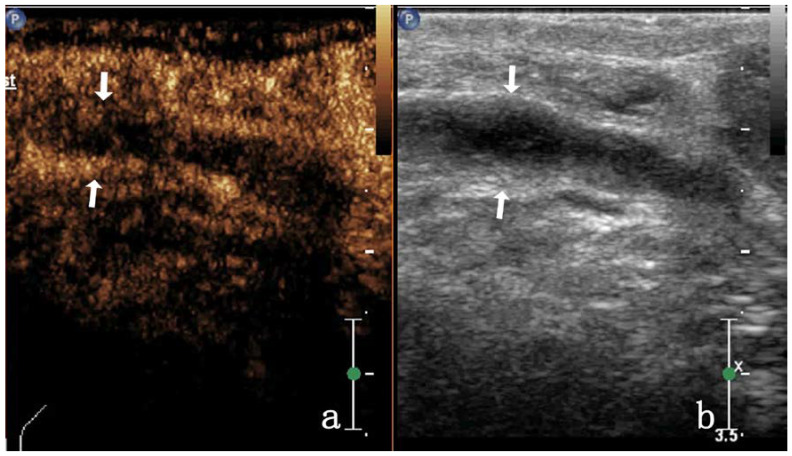
A 52-year-old man with VD TB. CEUS shows VD TB of low enhancement (arrow) in (**a**). 2D ultrasound of VD TB (arrow) in (**b**).

**Figure 5 diagnostics-13-01762-f005:**
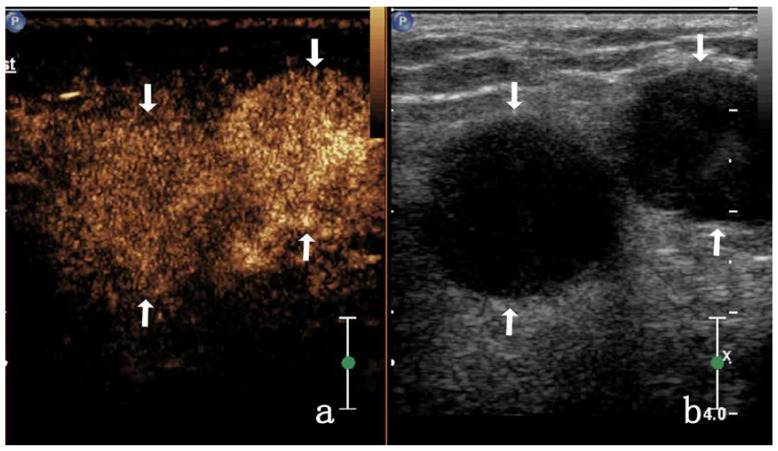
A 71-year-old man with inguinal MLN secondary to prostate cancer. CEUS shows lymph node homogeneous enhancement (arrow) in (**a**); 2D US shows hypoechoic lesions (arrow) in (**b**).

**Table 1 diagnostics-13-01762-t001:** Routine ultrasound findings of the two diseases.

Routine Ultrasound Manifestations		VD TB (17 Cases)	Metastatic Lymph Nodes (28 Cases)	χ^2^	*p*
Bilateral incidence		10/17 (58.82%)	16/28 (57.14%)	0.120	0.912
Unilateral onset		7/17 (41.17%)	12/28 (42.85%)		
Number of lesions	<3	6/17 (35.29%)	15/28 (53.57%)	0.947	0.331
	>3	11/17 (64.70%)	13/28 (46.42%)		
Maximum lesion size	<1 cm	5/17 (29.41%)	9/28 (32.14%)	NA	0.917
	1–3 cm	10/17 (58.82%)	14/28 (50.00%)		
	>3 cm	2/17 (11.76%)	5/28 (17.85%)		
Internal echogenicity	Hypoechoic	10/17 (58.82%)	18/28 (64.28%)	NA	0.080
	Isoechoic	1/17 (5.88%)	7/28 (25.00%)		
	Mixed echogenicity	6/17 (35.29%)	3/28 (10.71%)		
Internal calcification foci	+	3/17 (17.64%)	1/28 (3.57%)	NA	0.144
	−	14/17 (82.35%)	27/28 (96.42%)		
Lesions conglomeration	+	10/17 (58.82%)	6/28 (21.42%)	6.455	0.023
	−	7/17 (41.17%)	22/28 (78.57%)		
Sinus canal, skin rupture	+	0	2/28 (7.14%)	NA	0.519
	−	17/17 (100%)	26/28 (92.85%)		

**Table 2 diagnostics-13-01762-t002:** CEUS findings of the two diseases.

CEUS Enhanced Mode, Degree and Intensity		VD TB (17 Cases)	Metastatic Lymph Nodes (28 Cases)	χ^2^	*p*
CEUS degree				18.865	0.000
Homogeneous enhancement		3/17 (17.64%)	23/28 (82.14%)		
Heterogeneous enhancement	septal enhancement	4/17 (23.52%)	0		
	Annular enhancement	4/17 (23.52%)	5/28 (17.85%)		
	Nodule-in-nodule enhancement	3/17 (17.64%)	0		
non-enhancement		3/17 (17.64%)	0		
Enhanced intensity				17.455	0.000
	Low enhancement	5/17 (29.41%)	0		
	Equal enhancement	4/17 (23.52%)	0		
	High enhancement	8/17 (47.05%)	28/28 (100%)		
CEUS mode				15.074	0.000
Centripetal enhancement		0	16/28 (57.14%)		
Noncentripetal enhancement		17/17 (100%)	12/28 (42.85%)		

## Data Availability

All data materials in this manuscript can be applied to the corresponding author.

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
