# Peer review of "The Role of Contrast-Enhanced Ultrasound in the Differential Diagnosis of Tuberous Vas Deferens Tuberculosis and Metastatic Inguinal Lymph Nodes"

_diagnostics, 2023, doi:10.3390/diagnostics13101762_

Round 1

Reviewer 1 Report (New Reviewer)

I recommend only a few minor revisions of an editorial nature.

Some words has been written in red caracters: please control.

It is usefull to describle, in "pourpose" section, if it is an observational, prospective or retrospective study.

Abstract, line 2: there are two spaces beetween the words "(MLN)" and "in rutine": to be correct.

Keyword: ok

Main text

Line 27: insert a space beetwen these words "deferens(VD)"

Line 96: insert a space beetwen these words: "probe(If"

Line 136: insert space beetwen these words: "100%)(Figure 5)"

Figure: yuo can use abreviations for CEUS and other recurrent terms (for example in fig. 4 a)

Dicslaimer: nothing to say

Author Response

Some words has been written in red caracters: please control.

Reply :Thank you very much!

It is usefull to describle, in "pourpose" section, if it is an observational, prospective or retrospective study.

Reply :Thank you very much for the comments of the reviewers. I have added " retrospective analysis and summary …" in the purpose part.

Abstract, line 2: there are two spaces beetween the words "(MLN)" and "in rutine": to be correct.

Reply :Thank you very much for the careful review by the reviewers. I have deleted the blanks as required .

Keyword: ok

Reply :Thank you very much!

Line 27: insert a space beetwen these words "deferens(VD)"

Reply :Thank you very much for the careful review of the manuscript experts, I have added the space as required.

Line 96: insert a space beetwen these words: "probe(If"

Reply :Thank you very much for the careful review of the manuscript experts, I have added the space as required.

Line 136: insert space beetwen these words: "100%)(Figure 5)"

Reply :Thank you very much for the careful review of the manuscript experts, I have added the space as required.

Figure: yuo can use abreviations for CEUS and other recurrent terms (for example in fig. 4 a)

Reply :Thank you very much for the opinions of the reviewers. I have modified the description of the picture as required.

Dicslaimer: nothing to say

Reply :Thank you very much!

Reviewer 2 Report (New Reviewer)

Authors present a manuscript on the role of CEUS in differentiating VD TB and metastatic inguinal masses. Although the sample size is low (we should also consider the rarity of the diseases) some interesting results are presented. Such as the difference in homo/heterogeneity and centripetal enhancement.

Major comments

In my opinion, the paper would be more complete if the authors can provide also quantitative parameters of CEUS, which may add significative strength to this paper. 

Please describe the limitations of this study in the discussion

Minor comments

One more suggestion in the introduction authors state: At present, CEUS has been applied to liver, kidney, thyroid, breast and lymph nodes. The application of CEUS in deferens tuberculosis is rarely reported.

Please add citations for this statements. 

You can consult 

- Contrast-Enhanced Ultrasound (CEUS) in the Evaluation of Renal Masses with Histopathological Validation-Results from a Prospective Single-Center Study. Diagnostics (Basel). 2022 May 12;12(5):1209. doi: 10.3390/diagnostics12051209. 

- Performance of contrast-enhanced ultrasound (CEUS) in assessing thyroid nodules: a systematic review and meta-analysis using histological standard of reference. Radiol Med. 2020 Apr;125(4):406-415. doi: 10.1007/s11547-019-01129-2. 

Author Response

In my opinion, the paper would be more complete if the authors can provide also quantitative parameters of CEUS, which may add significative strength to this paper.

Reply :Thank you very much for the advice of the reviewers, which is very important for our subsequent research. In the future, we will compare the arrival time, peak time and enhancement duration of the contrast agents of the two diseases. We hope to invite you to review the manuscript again, thank you very much!

Please describe the limitations of this study in the discussion

Reply :Thank you very much for the comments of the reviewers. I have added the limitations of the research. Thank you again!

One more suggestion in the introduction authors state: At present, CEUS has been applied to liver, kidney, thyroid, breast and lymph nodes. The application of CEUS in deferens tuberculosis is rarely reported.

Please add citations for this statements.

Reply :I have added the references, thanks again!

This manuscript is a resubmission of an earlier submission. The following is a list of the peer review reports and author responses from that submission.

Round 1

Reviewer 1 Report

The authors present a manuscript that seeks to describe and differentiate tuberous vas deferens tuberulosis and metastatic inguinal lymph node.  The authors evaluate patients with pathologically confirmed inguinal nodes of tubrous vans deferens TB and compare the findings of routine ultrasound and contrast enhanced ultrasound, The exclude patiemts with advanced tuberous vas deferens TB as this is easy to distinguish from metastatic malignant TB. The authors provide decriptions of the ultrasound findings,

A number of specific issues of the manuscript include:

1. The first srntence in the abstract is somewhat misleading as the paper is describing the difference between not all vas deferens TB and metastatic inguinal nodes but tuberous vas defens TB (and even the only the early and probably mid stages but excluding the advanced stages) please correct this.

2, the conclusion in the abstract the CEUS revealed further imaging information not provided with routine ultasound- this point must come out here already

3. The third sentence of the introduction (line 29_ it is lomger it is not clear what they are comparing could the authors please clarify

4. in line 35 and 36 the is missing space after the comma and groin VD and spermatic cord VD

5. The vas deferens is in males what do the authors mean by can be also caused by menstrual infection in line 38 please could they please explain.

6. The authors introduce tuberous vas deferens TB but hardly give any explanation as to what that entity is, Is it the sonographuc finding described in reference one as one of the abnormal findings on sonographic imaging making this a sonographic rather that a clinical entity. Authors need to shed more light and this has implication on the subsequent discussion.

7. The authors report the incidence of tuberous vas deferens is not high do the authors have any data to support this statement, Again is the incidence low compared to other forms of vasx deferens TB

8. In linw 43 after talking about the low incidence of tuberous vas deferens TB r=that may lead to a misdiagnosis the authors begin to talk about metastatic inguinal nodes which id followed up in the discussion. Other authours who studied US findings and US use often discussed netastatic inguinal nodes with reference to a particular cancer such as vulvar. If as they authors did in this paper the want to discuss the use of ulrasoung in differential diagnosis the must not limit themselves to only metastatic inguinal nodes but also ,alignant nodes like that in lymphoma or non malignant but enlaged nodes that nay occur in infectius disease, The authors would otherwise have to explain why they limit themselves to differntiating the tuberous vas defens TB frim makignant nides.

9. Under methods inclusion- the word both before US in the phrase who underwent US and CEUS may nake it easiar to the reading audience

10. At what stage in the diagnosis were these patients included in the study- before any treatment was given?

11. The exlusion criteria already remove certain patients based on some morpological features seen on ultrasound- dose this represent most common presentation and is there a real clinical problem wouldnt such patient already have a clinical dignosis such as either TB of the genitourinal system or a makignancy and thus the US role would not be to differntiate one disease from another?

12. In order for this study to be replicated, the authors must elaborate on range of motion was determined by squeezing the lesion using the probe i.e. what cosituted no or limited motion and mobile

13, results in tables one and 2 the authors present number of patients fside by side for tuberous VD TB and metastatic lymph nodeswith the different total at the top to make the tables more presentable the precentage of the total must be at the side of the number so a comparison becomes standardized ie homogeneous enhancement 3/17 (17,6%)  23/28( (82,1%) so at a glance it is clear more patents had homogeneous enhancement using percentage i n the current way the figures are set up it dosent do justice to the presentation of the results.

14. in line 131 the percetage after 3/17 was omitted in the written results

15. The phrase a rare tuberculous pattern with strong echogenicity (line 176) seem misplaced please check.

16. The discussion is on the role in ,aking a differntial diagnosis with US in this context where the clinicsl history would likelybe known before evaaluation of nodes has been questiond under point 11 above perhaps consider the discussion in a different context to make the study more relevant to the clinician

17, in the results the age difference between tuberous VD TB and metastatic inguinal nodes was vast can a differentiation be made based on the age

Thank you very much

Reviewer 2 Report

I have read with great attention and interest the article entitled "The Role of Contrast-Enhanced Ultrasound in the Differential Diagnosis of Tuberous Vas Deferens Tuberculosis and Metastatic Inguinal Lymph Nodes"

Although the topic chosen by the authors is somewhat worthy of attention, the article has some serious flaws that prevent it from being published:

- The introduction is too long, wordy, generic and not related to the purpose of the study

- The methods are not clear and no outcomes are reported

- The results are really poorly described

- The discussion is too long

- The figures are of too low quality